

# The fate of fixed nitrogen in Santa Barbara Basin sediments during seasonal anoxia

Xuefeng Peng[1,2,3], David J. Yousavich[4], Annie Bourbonnais[1], Frank Wenzhöfer[5,6,7], Felix Janssen[5,6], Tina Treude[4,8] and David L. Valentine[2,3]

[1]School of Earth, Ocean and Environment, University of South Carolina, 701 Sumter Street, Columbia, SC, USA

[2]Marine Science Institute, University of California, Santa Barbara, CA, USA

[3]Department of Earth Science, 1006 Webb Hall, University of California, Santa Barbara, CA, USA

[4]Department of Earth, Planetary, and Space Sciences, University of California Los Angeles, 595 Charles E. Young Drive East, Los Angeles, CA, USA

[5]HGF-MPG Joint Research Group for Deep-Sea Ecology and Technology, Alfred-Wegener-Institute, Helmholtz Centre for Polar and Marine Research, Am Handelshafen 12, Bremerhaven, Germany

[6]Max Planck Institute for Marine Microbiology, Celsiusstrasse 1, Bremen, Germany

[7]Department of Biology, DIAS, Nordcee and HADAL Centres, University of Southern Denmark, Odense M, Denmark

[8]Department of Atmospheric and Oceanic Sciences, University of California Los Angeles, Math Science Building, 520 Portola Plaza, Los Angeles, CA, USA.

*Correspondence to*: Xuefeng Peng (xpeng@seoe.sc.edu) and David L. Valentine (valentine@ucsb.edu)





**Abstract.**

Despite long-standing interests in the biogeochemistry of the Santa Barbara Basin (SBB), there are no direct rate measurements of different nitrogen transformation processes. We investigated benthic nitrogen cycling using in-situ incubations with $^{15}NO_3^-$ addition and quantified the rates of total nitrate ($NO_3^-$) uptake, denitrification, anaerobic ammonia oxidation (anammox), $N_2O$ production, and 25 dissimilatory nitrate reduction to ammonia (DNRA). Denitrification was the dominant $NO_3^-$ reduction process, while anammox contributed 0 - 27% to total $NO_3^-$ reduction. DNRA accounted for less than half of $NO_3^-$ reduction except at the deepest station at the center of the SBB where $NO_3^-$ concentration was lowest. $NO_3^-$ availability and sediment total organic carbon content appeared to be two key controls on the relative importance of DNRA. The negative correlation between $NO_3^-$ availability and the 30 relative importance of DNRA suggests a negative feedback loop that potentially contributes to stabilizing the fixed N budget in the SBB. Nitrous oxide ($N_2O$) production as a fraction of total $NO_3^-$ reduction ranged from 0.2% to 1.5%, which was higher than previous reports from nearby borderland basins. A large fraction of $NO_3^-$ uptake was unaccounted for by $NO_3^-$ reduction processes, suggesting that intracellular storage may play an important role. Our results indicate that the SBB acts as a strong 35 sink for fixed nitrogen and potentially a net source of $N_2O$ to the water column.





# 1 Introduction

Oxygen minimum zones (OMZs) in the world's ocean, whether they are formed naturally or induced by human activities, have been expanding in the past century (Horak et al., 2016; Oschlies et al., 2017; Stramma et al., 2008). As oxygen ($O_2$) concentration is one of the key controls on biogeochemical processes, including nitrogen (N) cycling, N biogeochemistry in OMZs has been extensively studied (Paulmier and Ruiz-Pino, 2009; Zehr, 2009). Denitrification, the reduction of nitrate ($NO_3^-$) to dinitrogen gas ($N_2$), and anaerobic ammonia oxidation (anammox), where nitrite ($NO_2^-$) and ammonium ($NH_4^+$) are converted into $N_2$ by comproportionation are two major sinks of the oceanic fixed N budget (Gruber, 2008). These two processes are inhibited by the presence of $O_2$ and sulfide, and their rates are sensitive to $O_2$ at nanomolar concentrations (Dalsgaard et al., 2014; Joye and Hollibaugh, 1995; Caffrey et al., 2019). Because the last step of the sequential reduction of $NO_3^-$ during denitrification, $N_2O$ reduction, is the most sensitive to $O_2$ (Zumft, 1997), the production of nitrous oxide ($N_2O$) as a byproduct of denitrification is usually elevated under hypoxic conditions, i.e., in the presence of $O_2$ (Firestone et al., 1980; Ji et al., 2015). Additionally, nitrification, i.e., the oxidation of $NH_4^+$ and subsequently $NO_2^-$ is another major source of $N_2O$ in the ocean (Elkins et al., 1978), and the relative yield of $N_2O$ from nitrification is high under low-$O_2$ conditions (<4 μM) (Ji et al., 2018). Under $O_2$ limitation, dissimilatory nitrate reduction to ammonia (DNRA) coupled to organic matter degradation is another important process that results in fixed N retention instead of removal (Burgin and Hamilton, 2007). When viewed as competing processes, DNRA is favored over denitrification under $NO_3^-$-limited conditions where electron donors are in excess (Tiedje et al., 1983). Additionally, under sulfidic conditions, autotrophic DNRA coupled to sulfide oxidation can become a dominant pathway for $NO_3^-$ reduction (Shao et al., 2011, p.201).

The Santa Barbara Basin (SBB) is one of the borderland basins off the southern part of the coast of California and characterized by high export production (Thunell, 1998). Because the bottom water (maximum depth 586 m) in the SBB is separated from the area outside the basin by relatively shallow sills on the eastern end (~200 m deep) and the western end (~475 m deep), $O_2$ concentrations at the basin's bottom is generally low and usually fluctuates between 1 and 30 μM (Bograd et al., 2002;



Goericke et al., 2015; Reimers et al., 1990; Sholkovitz and Gieskes, 1971; Myhre et al., 2018). During upwelling seasons (winter and spring), water is advected from outside the basin and replenishes bottom water $O_2$ in the SBB. However, high export production fuels $O_2$ demand that maintains low $O_2$ levels within the basin at depths below the deeper sill (Thunell, 1998). As a consequence, anoxia develops at

the bottom of the SBB until the next upwelling event (Goericke et al., 2015).

Using water column $NO_3^-$ concentration data collected in the SBB by the California Cooperative Oceanic Fisheries Investigations (CalCOFI) along longitudinal transects (Koslow et al., 2010), Valentine et al. (2016) estimated the benthic $NO_3^-$ uptake rate to be as high as 11.7 mmol $m^{-2}$ $d^{-1}$, which

was one of the highest rates ever reported. However, the fate of the $NO_3^-$ in the sediments remains unclear as there are no direct rate measurements of N cycling processes in the SBB. Indirect estimates using analysis of stable isotopes of water column $NO_3^-$ suggests that benthic denitrification accounts for > 75% of $NO_3^-$ loss in the SBB, and the rates of benthic denitrification were estimated to be the highest among borderland basins in the eastern tropical North Pacific (Sigman et al., 2003). With respect to

$N_2O$, these other borderland basins are considered to be a weak sink (Townsend-Small et al., 2014). As the SBB stands out in terms of denitrification, it may be expected that SBB benthic cycling of $N_2O$ is also unique. Benthic anammox is expected to occur in the SBB (Prokopenko et al., 2006), but the relative contribution of denitrification and anammox to $N_2$ production has not been assessed.

To decipher the fate of $NO_3^-$ taken up by SBB sediments, we performed in-situ incubations using benthic flux chambers with added $^{15}NO_3^-$ along the bottom slope traversing north-south across the deeper portion of the SBB. By calculating the rates of $N_2$ production by denitrification and anammox, total $N_2O$ production, and DNRA, we assess the overall rates of $NO_3^-$ uptake and reduction rates. Accompanying geochemical data are used to explore the controls on the relative importance of $NO_3^-$

retention via DNRA.



## 2 Materials and Methods

### 2.1 In situ incubations with benthic flux chambers

Remotely operated vehicle (ROV) Jason deployed automated benthic flux chambers (BFC) and conducted sediment push coring at seven stations (Fig. 1) in the SBB along a southern and a northern depth and $O_2$ gradient originating from the depocenter in the deepest point of the basin (Table 1). Station depth, latitude, and longitude were automatically generated by the Jason data processor using navigation data derived from the Doppler Velocity Log system and the ultrashort baseline positioning system. Bottom water $O_2$ concentration was determined using a Type 4831 $O_2$ optode sensor (Aanderaa Data Instruments AS, Bergen, NO) on the ROV and calibrated against Winkler titration measurements of sweater collected from Niskin bottles (Qin et al., 2022). Bottom water was collected using Niskin bottles and stored frozen at -30°C until lab analysis for nitrate ($NO_3^-$) concentration following the spectrophotometric method described by (García-Robledo et al., 2014).

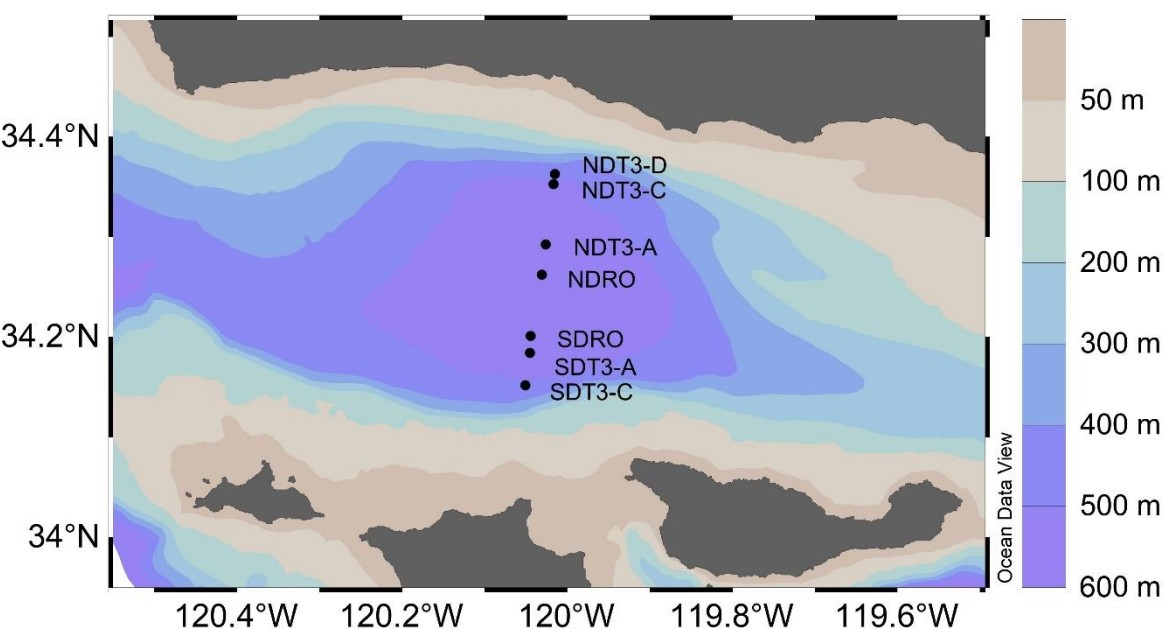

**Figure 1**. Sampling stations in the Santa Barbara Basin. The color contours show bathymetry data from the General Bathymetric Chart of the Oceans at 30 arc seconds resolution (Becker et al., 2009) visualized in Ocean Data View v5.6.2 (Schlitzer, 2002).



**Table 1.** Sampling date, latitude, longitude, depth, bottom water concentrations of oxygen and nitrate, chamber volume, total organic carbon (TOC) and nitrogen (TON), and C:N ratio of organic matter in the top 2 cm of the sediment (by dry weight %) at the seven sampling stations in the Santa Barbara Basin. Oxygen concentrations below detection limit of the Type 4831 (Aanderaa Data Instruments AS, Bergen, NO) oxygen optode sensor (3 μM) and the Winkler titration method (1 μM) is denoted by "bdl". Note that oxygen concentrations in the bottom water at NDRO and SDRO were confirmed to be zero through additional analytical methods (see Yousavich et al. 2023).

| Station | NDT3-D | NDT3-C | NDT3-A | NDRO | SDRO | SDT3-A | SDT3-C |
|---|---|---|---|---|---|---|---|
| Date | 7 Nov 2019 | 6 Nov 2019 | 4 Nov 2019 | 4 Nov 2019 | 3 Nov 2019 | 2 Nov 2019 | 8 Nov 2019 |
| Latitude | 34.363°N | 34.353°N | 34.292°N | 34.261°N | 34.201°N | 34.184°N | 34.152°N |
| Longitude | 120.015°W | 120.016°W | 120.026°W | 120.031°W | 120.045°W | 120.047°W | 120.050°W |
| Depth (m) | 447 | 498 | 572 | 580 | 586 | 571 | 494 |
| Chamber ID | BFC1 | BFC1 | BFC1 | BFC3 | BFC1 | BFC1 | BFC1 |
| Chamber volume (L) | 3.435 | 4.321 | 3.925 | 3.791 | 2.416 | 2.719 | 3.092 |
| Bottom water $O_2$ (μM) | 8.7 | 5.2 | 9.2 | bdl | bdl | bdl | 3.1 |
| Chamber $O_2$ (μM) at $T_0$ | 8.0 | 6.0 | 7.5 | 3.5 | 3.0 | 2.5 | 6.5 |
| Chamber $O_2$ (μM) at $T_{end}$ | 7.0 | 6.5 | 8.5 | 10.0 | 1.0 | 1.7 | 6.5 |
| Nitrate (μM) | 27.3 | 26.0 | 24.4 | 18.5 | 9.9 | 20.4 | 16.3 |
| TOC (%) | 4.1% | 4.6% | 5.9% | 5.7% | 6.2% | 6.8% | 5.3% |
| TON (%) | 0.5% | 0.5% | 0.7% | 0.7% | 0.8% | 0.9% | 0.6% |
| C:N ratio | 8.85 | 8.65 | 8.11 | 8.04 | 7.95 | 7.53 | 8.25 |

Sediment samples for total organic carbon (TOC) and total organic nitrogen (TON) analyses were subsampled from push cores (polycarbonate, 30.5 cm length, 6.35 cm inner diameter) retrieved by ROV Jason that were sectioned in 1-cm increments up to 10 cm followed by 2-cm increments below 10 cm (Yousavich et al., 2023). Wet sediments were dried for up to 48 hours at 50°C and treated with 6N HCl to dissolve carbonate minerals (Harris et al., 2001). Samples were then washed with ultrapure water and



dried again at 50°C. An aliquot (~10-15 mg) was then packed into individual 8x5 mm pressed tin capsules and analyzed at the University of California Davis stable isotope facility using a PDZ Europa

20-20 isotope ratio mass spectrometer (Sercon Ltd., Cheshire, UK). TOC and TON were calculated based on the sample peak area corrected against a reference material (alfalfa flour). Molar concentrations, obtained from measured TOC and TON (in wt%) were used to calculate carbon-to-nitrogen (C:N) ratios.

The design of the BFCs has been described previously (Vonnahme et al., 2020). In brief, a stirred cylindrical polycarbonate chamber (inner diameter = 19 cm) equipped with conductivity and oxygen sensors in the lid (type 5860 and 4330, respectively, Aanderaa Data Instruments AS, Bergen, NO) was inserted into the sediment to enclose a sediment patch of 284 cm$^2$ together with 2.5 to 4.5 L of overlying water. The chambers were outfitted with a syringe sampler hosting one injection syringe and six

sampling syringes to inject into and take samples from the overlying water at approximately 60-minute intervals. The injection syringe contained 200 µmol of $^{15}$N-labeled potassium nitrate (Cambridge Isotopes) dissolved in 50 ml of deionized water. To minimize the introduction of $O_2$, the $^{15}$N-labeled potassium nitrate solution was purged by ultra-high purity helium at 5 ml min$^{-1}$ for 60 minutes prior to be loaded into the injection syringe. The post-injection decrease in salinity in the chamber (as detected

by the conductivity sensor) was used to calculate the volume of the benthic flux chamber (Kononets et al., 2021). Depending on the chamber volume, the total concentration of $NO_3^-$ ranged between 50 and 100 µM at the beginning of in-situ incubations. This level of $NO_3^-$ amendment was intended to prevent its depletion before the end of incubations given the potentially high rates of $NO_3^-$ uptake estimated by a previous study (Valentine et al., 2016).


Water samples from the BFC were transferred to evacuated 12-ml vials (Exetainer®, Labco, Lampeter, UK) pre-filled with 0.1 ml of 7 M zinc chloride for preservation. Prior to analysis of the isotopic compositions of $N_2$ and $N_2O$, 5 mL sample was replaced with ultra-high purity helium to create a headspace. The concentration and $\delta^{15}N$ of dissolved $N_2$ and $N_2O$ was determined using a Sercon

CryoPrep gas concentration system interfaced to a Sercon 20-20 isotope-ratio mass spectrometer



(IRMS) at the University of California Davis Stable Isotope Facility. The measurement precision was ±0.2 ‰ for $\delta^{15}N$.

Water samples from the benthic flux chambers for analysis of $^{15}NH_4^+$ were filtered through sterile 47-mm syringe filters (0.2 μm pore size) and frozen immediately. The production of $^{15}NH_4^+$ in seawater samples was measured using a method adapted from *Zhang et al.* (2007) and described previously (Peng et al., 2016). In brief, $NH_4^+$ was first oxidized to $NO_2^-$ using hypobromite ($BrO^-$) and then reduced to $N_2O$ using an acetic acid-azide working solution (McIlvin and Altabet, 2005; Zhang et al., 2007). The $\delta^{15}N$ of the produced $N_2O$ was determined using an Elementar Americas PrecisION continuous flow, multicollector, isotope-ratio mass spectrometer (CF-MC-IRMS) coupled to a custom-built automated gas extraction and preparation system similar to the system described in McIlvin and Casciotti (2011). Calibration and correction were performed as described in Zhang et al. (2007). The measurement precision was ± 0.2‰ for $\delta^{15}N$. $NH_4^+$ solutions (10 μM) from a mixture of 99% $^{15}NH_4Cl$ (Cambridge Isotopes) and IAEA standard N1 ($\delta^{15}N$ = 1.2‰) with a final $\delta^{15}N$ of 135‰, 676‰, 1,351‰, 5,404‰, and 10,806‰ were prepared and used as in-house reference standards. The IRMS measurements of these in-house reference standards scaled linearly ($R^2$ = 0.9996) with their $\delta^{15}N$ values.

## 2.2 Rate calculations and statistics

Production rates of $^{29}N_2$, $^{30}N_2$, $^{15}NH_4^+$, and total $N_2O$ were calculated from the slope of the concentrations of the respective species at the syringe sampling time points by fitting a linear regression multiplied by the overlying water column volume and divided by the chamber area. The linear regressions excluded the last one or two sampling time points if they clearly deviated from a linear trend compared to the first four or five sampling time points. The rates of $N_2$ production from denitrification and anammox were calculated following a previously described method (Thamdrup and Dalsgaard, 2002) with modifications to account for coupled DNRA-anammox. The calculation was set up with denitrification rate ($R_{DN}$) and anammox rate ($R_{AMX}$) as unknowns:

$$R_{DN} \cdot f_N^2 + R_{AMX} \cdot f_A \cdot f_N = P^{30} \qquad\qquad \text{(Eqn. 1)}$$



$$R_{DN} \cdot 2 \cdot f_N \cdot (1 - f_N) + R_{AMX} \cdot [f_A \cdot (1 - f_N) + (1 - f_A) \cdot f_N] = P^{29} \qquad \text{(Eqn. 2)}$$

where $P^{29}$ and $P^{30}$ are the respective production rates of $^{29}N_2$ and $^{30}N_2$ that were calculated from measured concentrations stated above, $f_N$ is the fraction of $^{15}N$ in the $NO_3^-$ pool and $f_A$ is the fraction of $^{15}N$ in the $NH_4^+$ pool. The solution for $R_{DN}$ and $R_{AMX}$ is:

$$R_{DN} = \frac{(f_A + f_N - 2 \cdot f_A \cdot f_N) \cdot P^{30} - f_A \cdot f_N \cdot P^{29}}{f_N^2 \cdot (f_N - f_A)} \qquad \text{(Eqn. 3)}$$

$$R_{AMX} = \frac{f_N \cdot P^{29} - 2 \cdot (1 - f_N) \cdot P^{30}}{f_N \cdot (f_N - f_A)} \qquad \text{(Eqn. 4)}$$

Errors calculated from the linear regression of $^{29}N_2$ and $^{30}N_2$ production rates were propagated to $R_{DN}$ and $R_{AMX}$ following established statistical methods (Deming, 1943). Detection limits of the calculated rates were estimated as double the standard deviation from linear regressions. Depending on the in-situ $NO_3^-$ concentration, the detection limit for total $N_2$ production from denitrification and anammox ranged between 0.04 and 0.17 mmol m$^{-2}$ d$^{-1}$ and 0.04 and 0.24 mmol m$^{-2}$ d$^{-1}$ (Table S1), respectively. The detection limit for $N_2O$ production ranged between 1.1 and 5.6 µmol m$^{-2}$ d$^{-1}$. DNRA rates were calculated as the rates of increase in $^{15}NH_4^+$ divided by $f^{15}$, where $f^{15}$ is the fraction of $^{15}N$ in the $NO_3^-$ pool. Because part of the produced $^{15}NH_4^+$ would be adsorbed to sediment minerals, the rates of $^{15}NH_4^+$ production were further multiplied by a factor of two (De Brabandere et al., 2015; Laima, 1994). Depending on the in-situ $NH_4^+$ concentration, the detection limit for total $NH_4^+$ production rates ranged between 0.01 and 0.07 mmol m$^{-2}$ d$^{-1}$ (Table S1).

## 3 Results and Discussion

### 3.1 Interpretation of rate measurements from benthic flux chamber incubations

The use of benthic flux chambers to perform $^{15}NO_3^-$ incubation experiments in situ offers multiple advantages over other techniques such as slurry or whole-core incubations, including minimal disturbance of the sediment, maintenance of in-situ pressure and temperature, and relatively large surface area which can account for spatial heterogeneity (Aller et al., 1998; Hall et al., 2007; Nielsen and Glud, 1996; Robertson et al., 2019). One shortcoming of the tracer incubations with benthic flux chambers in this study is that the diffusion of added $^{15}NO_3^-$ into sediments and the labeled $^{15}NO_3^-$





reduction products out of sediments was unlikely at steady state. $^{15}NO_3^-$ added to the overlying water of the chambers diffuses into sediment porewater where $O_2$ is depleted within the first few millimeters,

sustaining benthic $NO_3^-$ reduction. However, a share of the labeled N-compounds that are produced will diffuse to pore waters in deeper sediment layers and, hence, cannot be detected in samples taken from the overlying waters. $O_2$ in bottom water (and, therefore, also in pore waters) was depleted (below detection of the Winkler titration method, 1 μM) at the deepest stations SDRO and NDRO (Table 1). The $NO_3^-$ reduction rates measured in our experiments represent only the benthic contribution because

the water samples in the six sampling syringes were sub-sampled simultaneously after recovery and no preservative was added inside the sampling syringe to terminate reactions. Therefore, we assume that $NO_3^-$ reduction in the overlying water contributed equally among all six sampling syringes to the production of $N_2$, $N_2O$, and $NH_4^+$, and does not interfere with our rate calculations. Separate water incubations would be needed to determine the rates of $NO_3^-$ reduction in the water column.


To account for $NH_4^+$ adsorption which could lead to an underestimate of DNRA, we made the assumption that an amount of $^{15}NH_4^+$ that equals the measured increase in the benthic flux chambers is adsorbed to sediment minerals (Hall et al., 2017; Laima, 1994). Although the reported production rates of $N_2$, $N_2O$, and $NH_4^+$ only accounted for the changes apparent in the chamber water column, they may

not be underestimates of the actual rates, because the addition of $NO_3^-$ at concentrations that were 1.6 - 6.2 (median = 2.3) times as high as ambient concentrations resulted in $NO_3^-$ uptake rates elevated by a factor of 1.9 - 6.4 (median = 3.8) as compared to those measured in parallel chambers deployed at the same time without any added substrates (Table S2; (Yousavich et al., 2023). While the diffusive loss of $NO_3^-$ to the sediment porewater is expected to account for the stimulated $NO_3^-$ uptake partially, $NO_3^-$

addition also likely stimulated the rates of $NO_3^-$ reduction and intracellular storage. However, it remains unclear whether the accelerated $NO_3^-$ uptake is partitioned between intracellular storage and reduction in the same proportion as under unamended conditions, which would partially depend on the carrying capacity of $NO_3^-$ storage vs. reduction. Therefore, we conservatively interpret the production rates of $N_2$, $N_2O$, and $NH_4^+$ reported here to be on the same order of magnitude as the actual rates, and the

relative contribution of different $NO_3^-$ reduction processes to be representative of in-situ conditions.



After all, the reported areal rates only represent benthic processes, which were estimated to account for three quarters of the denitrification in the SBB (Sigman et al., 2003).

$O_2$ concentrations in the overlying water in most incubations were slightly increasing over the time period of the incubation with an average rate of $0.11 \pm 0.44$ µmol $h^{-1}$. The increase is attributed to a release of $O_2$ from the polycarbonate walls and lids of the chambers that were exposed to air until shortly before deployment. The net increase in $O_2$ in the overlying water indicates that rates of $O_2$ provision from the plastics were in most cases higher than the rates of $O_2$ uptake by the enclosed sediment. A release of $O_2$ from plastics has been reported by a previous study which showed rates of $O_2$ provided from polycarbonate to $O_2$-poor waters were among the highest of all plastics tested (Stevens, 1992). The extent to which the artificial elevation of $O_2$ levels in the water overlaying the sediment in the chambers may have affected N-transformation pathways and rates will depend on the $O_2$ sensitivity of the respective processes and the penetration depth of $O_2$ into the sediment. This effect was likely insignificant in our incubations in the SBB because the rate of $O_2$ change was minimal compared to ambient $O_2$ concentrations except for station NDRO (Table 1). We note that at station NDRO, $O_2$ concentration in the chamber water rose from below detection to 10 µM, which likely resulted in an underestimate of the $NO_3^-$ reduction rates.

**3.2 Denitrification was the dominant $NO_3^-$ reduction pathway**

On average, $N_2$ production by denitrification and anammox was dominant over DNRA in this study, accounting for $70.4 \pm 16.4\%$ of total $NO_3^-$ reduction (Fig. 2 and Table 2). Total $N_2$ production rates ranged from 0.89 to 3.60 mmol N $m^{-2}$ $d^{-1}$, which were lower compared to a previous estimate (~4.5 mmol N $m^{-2}$ $d^{-1}$) based on $NO_3^-$ stable isotope mass balance calculations for the SBB (Sigman et al., 2003). Nevertheless, the previous estimate includes large uncertainties and the rates calculated from stable isotope mass balance represent signals integrated over multiple seasons (Sigman et al., 2003), whereas our measurements represent snapshots obtained in one season of one year when the bottom water $NO_3^-$ was not depleted. $N_2$ production rates at seasons more depleted in $NO_3^-$ concentrations in the bottom water compared to our study might more closely resemble rates estimated by Sigman et al.





(2003). Season-resolving studies are needed in the future to understand the natural variability of the system and assess potential effects of stressors such as deoxygenation and rising temperature.

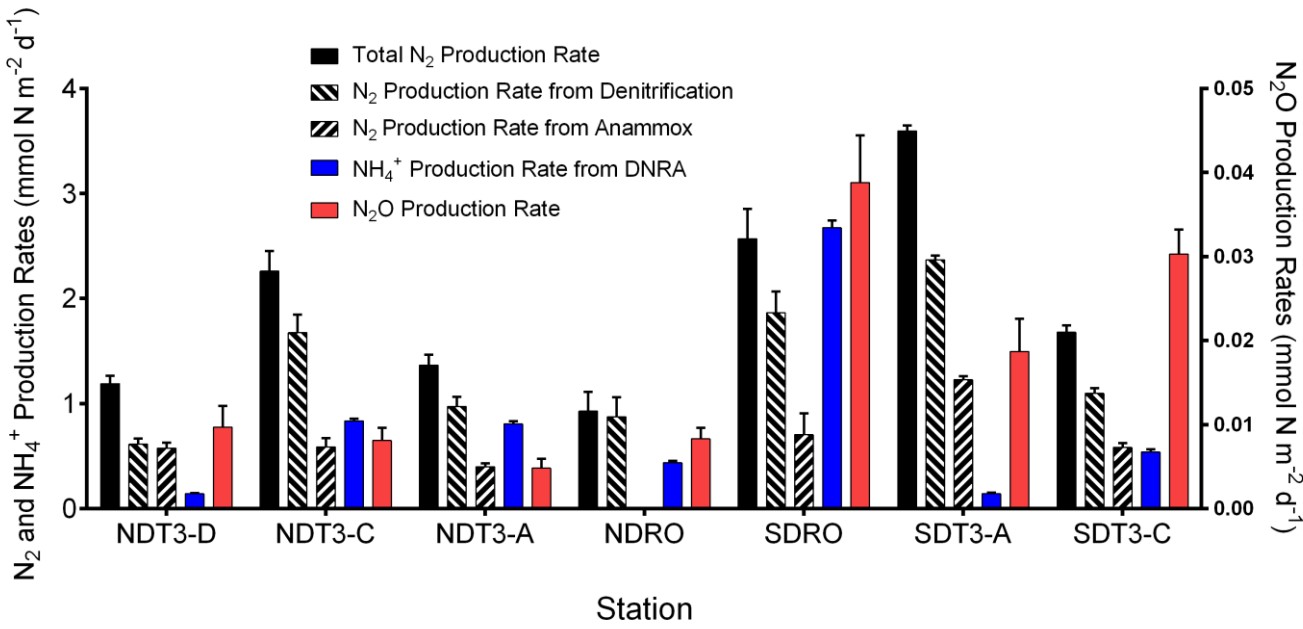


**Figure 2.** Inorganic N-species production rates determined from $^{15}N-NO_3^-$ labelling studies with in-situ benthic flux chambers: Production of total $N_2$, $N_2$ from denitrification, $N_2$ from anaerobic ammonia oxidation (anammox), $NH_4^+$ from dissimilatory nitrate reduction to ammonia (DNRA), and $N_2O$. Note the lower range (right y-axis) for $N_2O$ production.


$N_2$ production rates in this study were higher than most of those reported in other studies using in-situ incubations with benthic flux chambers (Bonaglia et al., 2017; De Brabandere et al., 2015; Hall et al., 2017; van Helmond et al., 2020; Hylén et al., 2022). Elevated rates in the SBB are likely a result of the high organic matter content of sediment (4.1 - 6.8% total organic carbon; Table 1), supporting high

microbial respiration rates, and little (max. 20 mm) to zero $O_2$ penetration into the sediment (Yousavich et al., 2023). Compared to the SBB, organic matter content in sediment of previous studies, including the anoxic Eastern Gotland Basin (Hall et al., 2017), the largely pristine and oxygenated Gulf of Bothnia (Bonaglia et al., 2017), and an anoxic fjord basin in the By Fjord on the Swedish west coast (De Brabandere et al., 2015), was lower and the $N_2$ production rates were typically < 1 mmol N m$^{-2}$ d$^{-1}$.



In comparison, $N_2$ production rates reached $1.72 \pm 0.77$ mmol N m$^{-2}$ d$^{-1}$ in the sediment underlying eutrophic waters of Stockholm archipelago, where organic matter content was similar to SBB sediment (6.3% w/w) and $O_2$ penetration depth was < 4 mm (van Helmond et al., 2020). Additionally, benthic denitrification rates in the SBB ($1.37 \pm 0.64$ mmol N m$^{-2}$ d$^{-1}$) were similar to those reported from the Peruvian OMZ ($1.31 \pm 0.60$ mmol N m$^{-2}$ d$^{-1}$) where bottom water $O_2$ was lower than 10 µM and the

organic matter content was similar (up to 7.5% TOC and 0.9% TON) to that in SBB sediments (Bohlen et al., 2011; Henrichs and Farrington, 1984; Sommer et al., 2016).

**Table 2.** The relative contribution of different processes (total $N_2$ production, $N_2$ from denitrification, $N_2$ from
anammox, $NH_4^+$ from DNRA, and $N_2O$ Production) to total $NO_3^-$ reduction (upper part) and the relative contribution of total $NO_3^-$ reduction to total $NO_3^-$ uptake (lower part) in the Santa Barbara Basin. Total $N_2$ production consists of $N_2$ from denitrification and $N_2$ from anammox. Total $NO_3^-$ reduction consists of total $N_2$ production, $NH_4^+$ from DNRA, and $N_2O$ Production. Total $NO_3^-$ uptake consists of total $NO_3^-$ reduction and other $NO_3^-$ sinks (e.g. intracellular storage).


| Processes contributing to Total $NO_3^-$ Reduction | NDT3-D | NDT3-C | NDT3-A | NDRO | SDRO | SDT3-A | SDT3-C |
|---|---|---|---|---|---|---|---|
| Total $N_2$ Production | 85.8% | 70.2% | 59.2% | 66.7% | 45.1% | 94.9% | 71.1% |
| $N_2$ from Denitrification | 59.2% | 60.4% | 49.3% | 66.7% | 38.3% | 75.8% | 56.8% |
| $N_2$ from Anammox | 26.6% | 9.8% | 9.9% | 0.0% | 6.8% | 19.1% | 14.3% |
| $NH_4^+$ from DNRA | 13.3% | 29.5% | 40.6% | 32.7% | 54.1% | 4.5% | 27.3% |
| $N_2O$ Production | 0.9% | 0.3% | 0.2% | 0.6% | 0.8% | 0.6% | 1.5% |
| *Total $NO_3^-$ Reduction* | 100% | 100% | 100% | 100% | 100% | 100% | 100% |
| **Processes contributing to Total $NO_3^-$ Uptake** | | | | | | | |
| Total $NO_3^-$ Reduction | 7.4% | 34.5% | 16.3% | 17.7% | 57.7% | 16.4% | 17.5% |
| Other $NO_3^-$ Sinks | 92.6% | 65.5% | 83.7% | 82.3% | 42.3% | 83.6% | 82.5% |
| *Total $NO_3^-$ uptake* | 100% | 100% | 100% | 100% | 100% | 100% | 100% |





Benthic denitrification rates exceeded anammox rates at all sampling sites (Fig. 2 and Table 2). This relationship agrees with the paradigm that denitrification is typically favored over anammox in organic-rich sediments (Dalsgaard et al., 2005; Devol, 2015). Anammox bacteria can reduce $NO_3^-$ to $NO_2^-$, which is then used to oxidize ammonia ($NH_3$) to $N_2$ (Kartal et al., 2007). In our in-situ incubations, coupled DNRA-anammox in which DNRA produces a substrate ($NH_3$) required by anammox could result in the production of $^{30}N_2$ (Prokopenko et al., 2006), which is accounted for by our rate calculation method (detailed in section 2.2). However, because the porewater $NH_4^+$ concentration was high (> 100 µM), the fraction of $^{15}N$ in the $NH_4^+$ pool remained low (up to 2.1% after ~1 hour of incubation and up to 4.3% after 6 hours of incubation). Therefore, the contribution of anammox to $^{30}N_2$ production was below 2.0% (Table S3). Overall, anammox contributed up to 26.6% of $NO_3^-$ reduction in the SBB (Table 2), indicating that anammox was a significant process in benthic SBB N cycling. Because the N isotope fractionation during the reduction of nitrite ($NO_2^-$) to $N_2$ by anammox bacteria (+16.0 ± 4.5‰) is lower than that of denitrification used for isotope mass balance calculations (~25‰), anammox likely contributed to the lower-than-expected natural abundance $^{15}N$ enrichment in the SBB water column $NO_3^-$ pool previously measured (Brunner et al., 2013; Sigman et al., 2003). When $NO_3^-$ is not limiting, denitrification typically dominates as the denitrifier population has a shorter generation time than DNRA bacteria (Kraft et al., 2014).

### 3.3 $NO_3^-$ availability and TOC control the relative importance of DNRA

The contribution of DNRA to total $NO_3^-$ reduction was lower than denitrification at all stations except for the deepest station SDRO (Fig. 2), where $NH_4^+$ production by DNRA contributed more than half of the $NO_3^-$ reduction (Table 2). The relative contribution of DNRA to total $NO_3^-$ reduction was positively correlated with TOC in the top 2 cm of the sediment (Fig. 3a) and negatively correlated with bottom water $NO_3^-$ concentration (Fig. 3b). These trends are consistent with previous findings showing that DNRA tends to be favored in environments with high availability of electron donors such as organic carbon (Hardison et al., 2015; Kraft et al., 2014; Tiedje et al., 1983) and limited by $NO_3^-$ (van den Berg et al., 2015; Kessler et al., 2018; Peng et al., 2016). Another example where DNRA dominated under limited $NO_3^-$ availability is reported from measurements along a bottom water $O_2$ and $NO_3^-$ gradient




traversing the Peruvian OMZ (Bohlen et al., 2011). One explanation for the increasing importance of DNRA under $NO_3^-$-limited conditions is that the growth yields calculated per mol electron acceptor from DNRA (consumes eight electrons) is higher than from denitrification (consumes five electrons) despite the greater amount of free energy provided by denitrification than DNRA per mol of $NO_3^-$, which was demonstrated by bacterial cultures capable of denitrification and DNRA (Strohm et al., 2007).

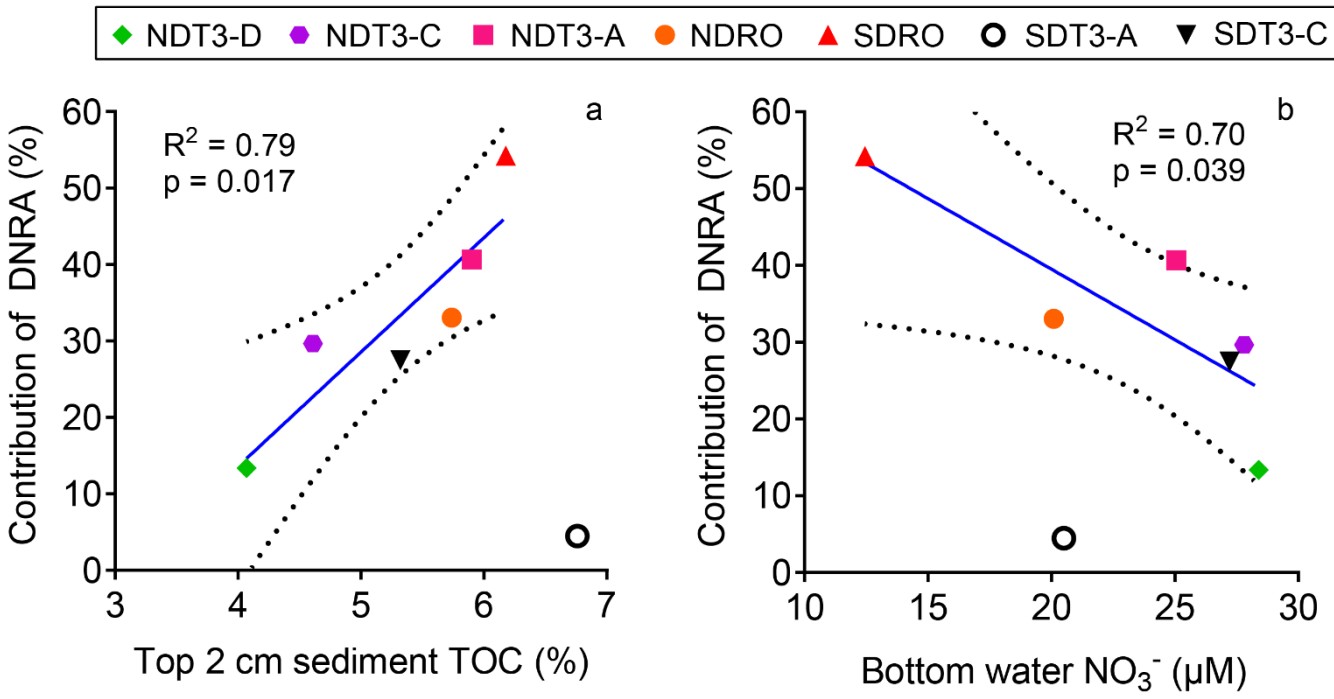

**Figure 3.** The correlation between the contribution of DNRA to $NO_3^-$ reduction (in %) and **(a)** the C:N ratio of sediment organic matter and **(b)** the bottom water $NO_3^-$ concentration in the Santa Barbara Basin. Linear regressions were performed excluding one outlier from station SDT3-A. The solid line represents the best fit, and the dashed lines represent the 95% confidence interval band.

The regressions we performed between the relative importance of DNRA vs. TOC and bottom water $NO_3^-$ concentration excluded one data point from the station SDT3-A that deviated from the overall





trend (Fig. 3). The DNRA rate at SDT3-A ($0.14 \pm 0.005$ mmol N m$^{-2}$ d$^{-1}$) was similarly low compared

to NDT3-D ($0.14 \pm 0.003$ mmol N m$^{-2}$ d$^{-1}$), but the N$_2$ production rates by both denitrification and anammox were the highest among all stations (Fig. 2), resulting in the lowest relative importance of DNRA. Porewater sulfide concentration was high at SDT3-A (Yousavich et al., 2023), so the DNRA bacteria should not be limited by the availability of electron donors. Sediments at SDT3-A were characterized by the highest TOC and TON content among all sites (Table 1), which likely fueled the

highest rates of denitrification and anammox (Middelburg et al., 1996).

The frequency and magnitude of seasonal anoxia in the SBB has been increasing in the past four decades, which is expected to intensify fixed N loss and NO$_3^-$ deficit in the water column (Goericke et al., 2015). Time-series measurements of water column NO$_3^-$ revealed that bottom water NO$_3^-$ depletion

has become more frequent since 2003 compared to the time between 1986 and 2003. While seasonal flushing of the SBB not only oxygenates the bottom water but also increases bottom water NO$_3^-$, our results suggest that fixed N retention via DNRA will increase in response to NO$_3^-$ drawdown even before NO$_3^-$ is near depletion, which effectively forms negative feedback that could potentially prevent the depletion of fixed N in the SBB. On the other hand, when NO$_3^-$ is no longer limiting, perhaps due to

slowdown of bottom water deoxygenation, the relative importance of DNRA would decrease, allowing denitrification to dominate NO$_3^-$ reduction pathways.

### 3.4 N$_2$O production and saturation

N$_2$O production rates measured by in-situ chamber incubations ranged from $4.8 \pm 1.1$ to $38.8 \pm 5.6$ µmol m$^{-2}$ d$^{-1}$ (Fig. 2). These rates were up to an order of magnitude higher than those measured using

shipboard whole-core incubations ($3.5 \pm 1.0$ µmol m$^{-2}$ d$^{-1}$) with samples from a similar depth (544 m) in the anoxic part of the Soledad Basin (Townsend-Small et al., 2014). A recent study using in-situ chamber incubations with $^{15}$NO$_3^-$ in the Eastern Gotland Basin reported rates (~15 - 68 µmol m$^{-2}$ d$^{-1}$) similar to or higher than the rates we measured in the SBB (Hylén et al., 2022). Because the physicochemical context of the Soledad Basin is more similar to the SBB than the Eastern Gotland

Basin, we expected the N$_2$O production rates in the Soledad Basin to be close to those in the SBB. The



much lower $N_2O$ production rates reported from the Soledad Basin may be partially attributed to the whole-core incubations that were not performed in situ.

$N_2O$ production as a fraction of total $NO_3^-$ reduction ranged from 0.2% to 1.5% (Table 2), which fell in
the typical range of $N_2O$ yield from both nitrification and denitrification (Ji et al., 2015, 2018). Although our measurements do not allow the distinction between $N_2O$ production from nitrification and denitrification, it is likely that both processes contributed with the respective share depending on ambient $O_2$ concentration. At the deepest stations where bottom water $O_2$ was depleted (Table 1), denitrification was likely the main source of $N_2O$. At other stations, where bottom water $O_2$ ranged from
3.1 - 9.2 µM, nitrification likely also contributed to $N_2O$ production.

Although we observed $N_2O$ production in all in-situ $^{15}NO_3^-$ incubations, $N_2O$ concentration in the chambers at the start of the incubations was far below saturation level (9 - 12%) at the two deepest stations SDRO and NDRO (Table S4). In contrast, $N_2O$ was either close to or above saturation at all
other stations (Table S4). The low concentration of dissolved $N_2O$ at the two deepest stations is consistent with our finding that $N_2$ production (i.e. $N_2O$ consumption) rates by denitrification were the highest there (Fig. 2), indicating that the deepest part of the SBB typically acts as a sink for $N_2O$. The shallower parts of the SBB were characterized by a lower $NO_3^-$ uptake rate (Table S2; Fig. S1), but they had a stronger potential for $N_2O$ production than the deepest stations (Fig. S2). In case of a
eutrophication event, enhanced surface primary productivity could stimulate denitrification as well as $N_2O$ production in the shallower parts of the SBB where bottom water $O_2$ is not depleted, where benthic $N_2O$ production is more likely to contribute to $N_2O$ efflux from the water column during upwelling events.

### 3.5 Total $NO_3^-$ uptake suggests high potential for intracellular $NO_3^-$ storage

Although the $N_2$ production rates we measured in the SBB were among the highest reported values for any marine sediments, total $NO_3^-$ reduction, which also includes DNRA and $N_2O$ production, only accounted for 23.9 ± 16.9% of the total $NO_3^-$ uptake in benthic flux chambers amended with $^{15}NO_3^-$





(Table 2). Intracellular $NO_3^-$ storage by bacteria and microbial eukaryotes was likely responsible for the majority of the $NO_3^-$ uptake unaccounted for by the different $NO_3^-$ reduction pathways. Marine
*Beggiatoa spp.* can hyper-accumulate $NO_3^-$ intracellularly at concentrations 3,000- to 4,000-fold above ambient levels (McHatton et al., 1996). Other microbial lineages including *Thioploca*, foraminifera, and gromiida are also known to store $NO_3^-$ intracellularly (Piña-Ochoa et al., 2010; Zopfi et al., 2001). In two of the porewater profiles sampled during the same cruise, $NO_3^-$ concentrations at 1 cm depth reached 80 - 390 μM, which we interpreted as evidence of $NO_3^-$ leakage from bacterial cells during
porewater handling (Yousavich et al., 2023). While it is difficult to directly constrain the contribution of intracellular $NO_3^-$ storage to total $NO_3^-$ uptake, it can be indirectly inferred by calculating the diffusive loss (both upward and downward) of added $^{15}NO_3^-$ if porewater concentrations in sediments underlying the benthic flux chamber were available.

The total $NO_3^-$ uptake in the SBB measured from parallel benthic flux chambers without substrate amendment at the same stations (3.26 ± 0.72 mmol N m$^{-2}$ d$^{-1}$) (Yousavich et al., 2023) was higher than that in other nearby borderland basins such as the San Nicolas Basin (0.38 ± 0.03 mmol N m$^{-2}$ d$^{-1}$), the San Pedro Basin (0.78 ± 0.11 mmol N m$^{-2}$ d$^{-1}$) (Berelson et al., 1987), and the Santa Monica Basin (1.10 ± 0.31 mmol N m$^{-2}$ d$^{-1}$) (Jahnke, 1990). As mentioned above (section 3.1), the addition of $^{15}NO_3^-$
stimulated $NO_3^-$ uptake rates by multiple folds (compared to BFC incubations without $^{15}NO_3^-$ additions) and to a level (11.60 ± 4.15 mmol N m$^{-2}$ d$^{-1}$) similar to a previous estimate (11.7 mmol N m$^{-2}$ d$^{-1}$) based on water column $NO_3^-$ deficit (Valentine et al., 2016). Since bottom water $NO_3^-$ during our sampling time (>12.5 μM in November 2019) was not as depleted as in October 2013 (~2 μM $NO_3^-$) (Valentine et al., 2016), these results indicate that the microbial community in SBB sediments have the metabolic
potential to further consume $NO_3^-$ when SBB bottom water undergoes extended periods (months) of anoxia during autumn and winter. Assuming that $NO_3^-$ in the lowermost 10 m of the water column are under direct influence of benthic $NO_3^-$ uptake, we estimate it would take between one to four months to deplete bottom water $NO_3^-$ with a starting concentration of 30 μM, with the shortest depletion time at the depocenter and the longest at the periphery of the SBB. This timescale agrees with time-series
measurements of water-column $NO_3^-$ concentrations in the SBB (Goericke et al., 2015), and it implies



that bottom water $NO_3^-$ is unlikely to become depleted at depths shallower than 500 m. Furthermore, we identified a significant negative correlation between $NO_3^-$ uptake rates without substrate amendments and the fold-change after $^{15}NO_3^-$ addition (Fig. S3). This negative correlation indicates that benthic $NO_3^-$ uptake rates at the shallow stations were the most responsive to exogenous $NO_3^-$ supply, while on the other hand $NO_3^-$ uptake rates at the deep and anoxic stations were closer to an upper limit that is determined by the microbial community present in the SBB sediments.

## 4 Summary

We investigated benthic nitrogen cycling processes using in-situ incubations with $^{15}NO_3^-$ addition and quantified the rates of total $NO_3^-$ uptake, denitrification, anammox, $N_2O$ production, and DNRA. Denitrification was the dominant $NO_3^-$ reduction process (38-76%), while anammox contributed up to 27%. DNRA accounted for less than half of $NO_3^-$ reduction except at the deepest station (586 m), at the center of the SBB, where bottom water $O_2$ concentrations were zero. The relative importance of DNRA was positively correlated with sediment TOC and negatively correlated with bottom water $NO_3^-$ availability. $N_2O$ production as a fraction of total $NO_3^-$ reduction ranged from 0.2% to 1.5%, which was higher than previous reports from nearby borderland basins. The large fraction of $NO_3^-$ uptake unaccounted for by $NO_3^-$ reduction processes suggests high potential for intracellular storage. Our results indicate the role of the SBB sediments as a strong sink for fixed nitrogen. Future intensification of water column anoxia may elevate the importance of fixed N retention via DNRA by keeping N in the system as $NH_4^+$, forming negative feedback that could overall reduce fixed N loss in the SBB.



## Data availability

The rate data in tabular form are available at
https://figshare.com/articles/dataset/Peng_et_al_2023_xlsx/21824610.

## Author contribution

XP, TT, and DLV designed the study; XP, DJY, FW, FJ, TT, and DLV participated in the fieldwork; XP, DJY, AB, FW, and FJ performed the measurements; XP wrote the manuscript; All authors contributed to the writing of the manuscript and discussion of the data.

## Competing interests

One of the co-authors is a member of the editorial board of Biogeosciences.

## Acknowledgements

We thank the captain, crew, and scientific party of the R/V Atlantis, and the crew of the ROV Jason for their technical and logistical support during the research expedition AT42-19. We also thank D. Robinson, S. Krause, Q. Qin, E. Arrington, M. O'Beirne, A. Mazariegos, X. Moreno, A. Eastman, H. Kittner, S. Dorji, J. Burgos-Ponce, N. Liu, J. Tarn, and K. Gosselin for assisting with shipboard analyses. Funding for this work was provided by the US National Science Foundation, NSF OCE-1756947 and OCE-1830033 (to DLV) and OCE-1829981 (to TT), and by a Simons Foundation Postdoctoral Fellowship in Marine Microbial Ecology (No. 547606 to XP).





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
