# Peer review of "The fate of fixed nitrogen in Santa Barbara Basin sediments during seasonal anoxia"

_EGUsphere, 2023_

## Author Comment (AC1)

**RC1**: 'Comment on egusphere-2023-1498', Anonymous Referee #1, 11 Mar 2024

This manuscript presents an interesting and detailed study on the benthic nitrogen cycling in the Santa Barbara Basin. The study is original as the authors used a complex approach measuring in-situ incubation, quantifying benthic rates of nitrate uptake, denitrification, anammox, nitrous oxide production, and DNRA. They found that the sediments in the Santa Barbara Basin acted a sinks for fixed nitrogen with dominating denitrification and as source for nitrous oxide. The data set is well presented and interpreted and the text well written and organized. I only have a few minor comments.

We thank the reviewer for the positive comments.

L141-142: How big is the effect of additional substrate added to chamber incubations? In-situ bottom water concentrations ranged from 9.9 µmol $L^{-1}$ to 27.3 µmol $L^{-1}$ (Table 1), after adding $^{15}N$-labeled nitrate concentration varied between 50 and 100 µmol $L^{-1}$. Do you suspect to overestimate rates to higher substrate availability?

Table S2 that is already included in the supplementary information shows the effect of additional substrate added to chamber incubations. Lines 220 - 223 acknowledges that:

"the addition of $NO_3^-$ at concentrations that were 1.6 - 6.2 (median = 2.3) times as high as ambient concentrations resulted in $NO_3^-$ uptake rates elevated by a factor of 1.9 - 6.4 (median = 3.8) as compared to those measured in parallel chambers deployed at the same time without any added substrates (Table S2; (Yousavich et al., 2024)."

Indeed, the additional substrate may have stimulated the nitrate reducing activities. On the other hand, because samples from benthic chamber incubations are taken from the overlying water they cannot provide detailed information about all processes that may occur in the underlying sediments. For example, we cannot be certain about the magnitude of nitrate reductions unaccounted for due to reduced products being adsorbed to sediments (e.g. ammonium). As a result, we cannot be sure about whether the reported nitrate reduction rates were overestimates or underestimates (Lines 216 - 219).

L256-257: How would rate changes with seasonal altering oxygen concentrations?

Nitrate reduction processes are inhibited by oxygen, although the oxygen sensitivity of different processes likely differ. Overall, we expect lower rates of nitrate reduction processes at higher oxygen concentrations. This was evident in the long-term monitoring dataset shown in Goericke et al., 2015, where nitrate deficits were correlated with the degree of anoxia in the Santa Barbara Basin.

L258-259: How representative are the results considering seasonal changes in oxygen and nitrate concentrations?

The results are representative of seasonal anoxia in the Santa Barbara Basin (SBB), which develops at least twice a year following upwelling events (winter and spring) (Goericke et al. 2015). Pronounced nitrate deficits accompany the anoxia due to nitrate reduction processes. This is explained in the Introduction in lines 62 – 70. In this study, we sampled during one time of the year, and we do not intend to use the results to represent seasonal changes in the SBB, which is now clarified in the updated version of the summary.

L297 "However, because the porewater $NH_4^+$ concentration was high […]": Have pore water or bottom water ambient ammonium concentrations been measured? I cannot find any information about porewater sampling in the method section. If you refer to another paper this statement needs a reference. Both anammox and nitrification, which according to the authors contributes at least in part to $N_2O$ production (L367), are dependent on available ammonium, it would be interesting to know the in-situ concentrations.

Thank you for the suggestion. We will include a reference after this statement at Line 297 (Yousavich et al. 2024), which is where porewater ammonium concentrations were published.

L364-370: Why do you not discuss the potential of DNRA to contribute to $N_2O$ production?

While there are reports of $N_2O$ production from bacteria capable of nitrate ammonification, none of the bacterial lineages are typically found in marine sediments. Bacillus vireti, Bacillus sp., and Citrobacter sp. were isolated from soil (Mania et al. 2014, Streminska et al. 2012), Bacillus licheniformis were isolated from silage, garden soil, and flour (Sun et al. 2016). Moreover, many of these $N_2O$-producing bacterial strains are capable of both DNRA and canonical denitrification, which confounds the distinction of $N_2O$ produced via denitrification vs. DNRA.

---

## Author Comment (AC2)

**RC2**: , Anonymous Referee #2, 15 Mar 2024

**General comment**

The authors investigated nitrogen cycling using *in-situ* incubations with the addition of $^{15}NO_3^-$ in the deep Santa Barbara basin, which is mainly anoxic. During incubations, the benthic uptake of total nitrate ($NO_3^-$), denitrification, anaerobic ammonium oxidation (anammox), dissimilatory nitrate reduction to ammonium (DNRA) and $N_2O$ production were assessed.

I agree that such incubations are challenging, but they provide new information on rates. In reality, the present contribution provides information about one more new study site using benthic chambers. At the same time, there are also many limitations to using the isotope pairing method during chamber incubations for studying sedimentary dissimilatory pathways. These limitations could be better addressed in the present study.

Thank you for the suggestion. The isotope pairing technique (IPT) was developed and primarily used with whole-core incubations, which comes with multiple challenges including: 1) bioturbation that can affect the $^{14}NO_3^-$ to $^{15}NO_3^-$ ratios within sediments; 2) maintaining low bottom-water oxygen concentrations; 3) gas ebullition that can disturb the redox zonation; and 4) the in situ bottom water flow field, and thus the corresponding porewater flow field, is not maintained (Robertson et al. 2019). Coupling the IPT to benthic chambers avoids some of these issues (No. 2 and 3) but not others (No. 1 and 4). While the IPT was developed to distinguish denitrification in the bottom water and in the sediments, in the present study, the aim is to determine the nitrate reduction processes in sediments only. The revised section 3.1 provides a more detailed discussion on the limitations associated with using $^{15}NO_3^-$ incubations with benthic chambers (see below).

Regarding benthic dissimilatory pathways, questions still need to be answered about whether the study underestimated $^{29/30}N_2$ and $^{15}NH_4^+$ production rates since samples were only collected from the water phase. Conventionally, $NO_3^-$ reduction rates are based on production rates in both the bottom and pore water phases. Additional explanations help address this point. Secondly, I am concerned about the time period that needs to reach diffusion equilibrium after the $^{15}NO_3^-$ addition.

Thank you for the suggestion. Given the methods used in this study, we focus on the reaction rates of benthic dissimilatory pathways and not in the bottom water. In whole-core IPT experiments, the equilibration period following $^{15}NO_3^-$ addition is important, but it only allows the determination of $NO_3^-$ reduction at *quasi*-steady state. In our manuscript, we already stated in the discussion that diffusion equilibrium may not have reached in our incubations. The time needed is hard to quantify as it will depend on the sediment depth where conversions are taking place – the closer to the sediment water interface the shorter the time period needed. In any case the general setup of the experiment with nitrate being supplied from the water column and should largely agree with conditions in the natural environment.

Also, I suggest a strongly revised discussion. At the present version of the manuscript, I could not recommend it for publication in BG. My specific and general comments are provided below.

Thank you for the suggestion. We have made extensive edits to the sections for which comments were made by any of the two reviewers.

**Specific comments**

Line 29-30: what do you mean here, "feedback loop"?

What is meant by this sentence is that as nitrate deficit intensifies, the relative importance of fixed N retention via DNRA increases, which should reduce the loss of fixed N (as nitrate from the sediment to bottom waters), and this can be considered a negative feedback loop. Conversely, when nitrate deficit is low, the relative importance of N retention via DNRA decreases, which would lead to an increase in fixed N loss.

Below is the edited sentence of this part of the abstract to clarify this meaning:

"The increasing importance of fixed N retention via DNRA relative to fixed N loss as $NO_3^-$ deficit intensifies suggests a negative feedback loop that potentially contributes to stabilizing the fixed N budget in the SBB."

Line 34-35: this is entirely speculative as the reader cannot see by which magnitude benthic processes could affect bottom/water concentrations of NO3 and N2O. The water column is quite high, and what proportion of the standing pool could be affected?

Previous hydrographic studies in the SBB have shown strong nitrate deficits in the water column in the bottom 100 m depths during intense anoxic events and it was facilitated by flushing of the SBB (Goericke et al. 2015). Our study has measured the high rates of nitrate drawdown, as well as most processes that contribute to this nitrate drawdown. Therefore, the first part of the last sentence in the abstract is not speculative.

There is an equivalent hydrographic record for water column $N_2O$ concentration. Although the $N_2O$ production quantified using benthic chambers in this study represent processes in the sediment, they were measured by sampling the water overlying the sediment. Therefore, our measurements directly represent how benthic processes affect bottom water concentrations of $N_2O$. Even though we do not argue the $N_2O$ produced from benthic processes could reach surface waters of the SBB, it is still possible given the nitrate deficit observed even in surface waters in the SBB during strong flushing events.

Line 57: what can lead to an increase in oxygen levels during incubations? Please clarify if the C: N ratio provided is molar. Consider adding subscripts for TOC and TON to understand better how they refer to sediments.

The increase in oxygen concentration during incubation was from the release of oxygen from the polycarbonate walls and lids of the benthic chambers, which is explained in Lines 235 – 245 of the original manuscript:

"The increase is attributed to a release of $O_2$ from the polycarbonate walls and lids of the chambers that were exposed to air until shortly before deployment. The net increase in $O_2$ in the overlying water indicates that rates of $O_2$ provision from the plastics were in most cases higher than the rates of $O_2$ uptake by the enclosed sediment. A release of $O_2$ from plastics has been reported by a previous study which showed rates of $O_2$ provided from polycarbonate to $O_2$-poor waters were among the highest of all plastics tested (Stevens, 1992). The extent to which the artificial elevation of $O_2$ levels in the water overlaying the sediment in the chambers may have affected N-transformation pathways and rates will depend on the $O_2$ sensitivity of the respective processes and the penetration depth of $O_2$ into the sediment. This effect was likely insignificant in our incubations in the SBB because the rate of $O_2$ change was minimal compared to ambient $O_2$ concentrations except for station NDRO (Table 1)."

Thank you for the suggestion. We have now calculated the C:N molar ratio and clarified it in the caption for Table 1. We have also added the word "Sediment" before "TOC", "TON", and "C:N ratio" in the first column of Table 1.

Line 85-90: the authors state what they did, which sounds quite descriptive. What was the motivation and aim of this study? I suggest providing questions that the authors aim to answer.

The motivation of this study was the previously reported high nitrate drawdown in the SBB (Goericke et al. 2015) and the large coverage of bacterial mats in the SBB (Valentine et al. 2016) (now included in the revised introduction). The aim of this study is to decipher the fate of nitrate taken up by SBB sediments. Both points here have been provided in the Introduction.

Line 139-144: please provide the final enrichment degree achieved. Maybe I am less familiar with chamber operation, but I need more explanation of how IPT assumptions were achieved during these chamber incubations (for e.g. D14 rates independence of 15N additions (I guess here authors used only D15 rates), equilibrium in diffusion etc.)

We will include the final enrichment degree as an additional row in Table S2 in the revised manuscript. While the manuscript includes multiple references to previous IPT studies, the present study does not attempt to distinguish denitrification rates in the water column (from ambient nitrate) vs. in sediments (from coupled nitrification-denitrification). This study used $^{15}NO_3^-$ incubations with benthic chambers to determine the rates of different dissimilatory nitrate reduction pathways. Therefore, many of the IPT assumptions are not relevant.

Line 169-171: here, the authors state that the production of $^{29/30}N_2$ was calculated from linear

regression; however, looking at the text related to the incubations, it seems they sampled once per incubation. Then, this is quite confusing.

The chamber was sampled at six different time points, in approximately 60 minutes intervals and measurements of those samples were used for the regression. Sampling is described in Lines 134 – 136:

"The chambers were outfitted with a syringe sampler hosting one injection syringe and six sampling syringes to inject into and take samples from the overlying water at approximately 60-minute intervals."

Line 197-232: section 3.1 aims to argue that the technique used did not underestimate measured rates. Overall, this section should be better developed.

Thank you for the suggestion. We have made significant revision to section 3.1 (included at the end of this document) to present a comprehensive and clear discussion of the limitations of our methods.

Lines 205-207: the authors recognize the weaknesses of the approach used. However, they need to identify what could potentially lead to underestimation.

Thank you for the suggestion. We have identified the two main limitations (oxygen introduction by the chamber plastics and loss of products from nitrate reduction to deeper sediment layers by downward diffusion) of our methods that could potentially lead to underestimation of true nitrate reduction rates. Please see the revised Section 3.1 (included at the end of this document).

Line 207: there appears to be a sudden jump in the logical flow.

Thank you for the suggestion. We have moved this sentence to improve the logical flow in the revised Section 3.1 (included at the end of this document).

Lines 219-223: need more concrete arguments regarding whether rates were underestimated.

As stated in section 3.1 we cannot make a final judgement as there are factors that may lead to over- and underestimation that could potentially balance each other out.

Line 222: the authors say that parallel incubations were performed, but I could not find this information in the Materials and Methods.

At the end of this sentence, we cited the study where parallel incubations were described, which is now published (Yousavich et al. 2024, Biogeosciences).

Line 225: the authors should have discussed intracellular storage in the introduction to clarify their meaning.

Thank you for the suggestion. In the revised manuscript, we will include an introduction to intracellular nitrate storage in the introduction.

Lines 297-299 need to be clarified on where the data presented comes from.

Thank you for the suggestion. In the revised manuscript we will cite Yousavich et al. (2024) as the source of the data.

Line 338-340:  this is quite a speculative statement.

This statement is supported by numerous previous studies that showed high organic carbon content of sediments is a primary driver for denitrification (e.g. Middelburg et al. 1996, Devol 2015). We will rephrase the sentence to allow for a greater degree of uncertainty:

"Sediments at SDT3-A were characterized by the highest TOC and TON content among all sites (Table 1). Consequently, highest rates of denitrification and anammox may be expected at these organic matter rich sites in agreement to what has been observed in other studies (Middelburg et al., 1996; Devol, 2015)."

Line 385-398: this is quite a speculative statement without supportive information.

Aside from intracellular storage of nitrate, we have considered and quantified all other possible fates of nitrate. A recently published companion paper has demonstrated that there are high levels of intracellular nitrate storage by bacteria in SBB sediments. This is stated in section 3.5 of the manuscript: "In two of the porewater profiles sampled during the same cruise, $NO_3^-$ concentrations at 1 cm depth reached 80 - 390 µM, which we interpreted as evidence of $NO_3^-$ leakage from bacterial cells during porewater handling (Yousavich et al., 2024)." Therefore, we do not consider this part of the discussion as highly speculative.

Line 423-434: in this section, authors should avoid the repetition of results.

Thank you for the suggestion. While we think it is necessary to highlight a few key results in the Summary section, we will revise the summary section to reduce repetition to the necessary minimum.

**Technical comments**

Table 1: The measures in the first column could be clearer, and it is challenging to understand which numbers are for the sediment or water column.

Thank you for the suggestion. We have now adjusted the positions of text in Table 1 so that the measures are clear. We have added "water column" before "nitrate" and "sediment" before "TOC", "TON", and "C:N ratio", which makes it clear which data are for the sediment or water column.

Figure 2 is quite complex and needs to be reshaped. I suggest providing the total N2 production and the partition between different dissimilatory pathways. Another option is to provide total nitrate reduction and partitioning between different pathways. I was surprised standard error bars are shown here without clarity on replication. Therefore, Table 2 could be modified.

Thank you for the suggestion. We have adopted the suggestion and plotted the total N2 production in two parts including N2 from denitrification and N2 from anammox (see figure below). To explain the error bars, we have now included in the caption of Figure 2 the following sentence:

[revised manuscript text omitted]